# A Comprehensive Review of Long Non-Coding RNAs in the Cancer–Immunity Cycle: Mechanisms and Therapeutic Implications

**DOI:** 10.3390/ijms26104821

**Published:** 2025-05-17

**Authors:** Mario Perez-Medina, Jesus J. Benito-Lopez, Dolores Aguilar-Cazares, Jose S. Lopez-Gonzalez

**Affiliations:** 1Laboratorio de Investigacion en Cancer Pulmonar, Departamento de Enfermedades Cronico-Degenerativas, Instituto Nacional de Enfermedades Respiratorias “Ismael Cosio Villegas”, Mexico City 14080, Mexico; mperezm1518@alumno.ipn.mx (M.P.-M.); jareb.benito.l@gmail.com (J.J.B.-L.); daguilarc@iner.gob.mx (D.A.-C.); 2Asociación Para Evitar la Ceguera en México, I. A. P., Mexico City 04030, Mexico

**Keywords:** long non-coding RNAs (lncRNAs), cancer–immunity cycle, immune evasion, immunotherapy resistance, tumor immunogenicity

## Abstract

Long non-coding RNAs (lncRNAs) have emerged as pivotal regulators of the dynamic interplay between cancer progression and immune responses. This review explored their influence on key processes of the cancer–immunity cycle, such as immune cell differentiation, antigen presentation, and tumor immunogenicity. By modulating tumor escape from the immune response, therapeutic resistance, and tumor–stroma interactions, lncRNAs actively shape the tumor microenvironment. Due to their growing knowledge in the area of immune suppression, directly intervening in the induction of regulatory T cells (Tregs), M2 macrophages, and regulating immune checkpoint pathways such as PD-L1, CTLA-4, and others, lncRNAs can be considered promising therapeutic targets. Advances in single-cell technologies and immunotherapy have significantly expanded our understanding of lncRNA-driven regulatory networks, paving the way for novel precision medicine approaches. Ultimately, we discussed how targeting lncRNAs could enhance cancer immunotherapy, offering new avenues for biomarker discovery and therapeutic intervention.

## 1. Introduction

Cancer remains one of the greatest challenges in modern medicine due to the ability of tumor cells to proliferate uncontrollably, evade cell death mechanisms, and develop resistance to treatment [1]. Within the tumor microenvironment (TME), tumor, stromal, and immune cells establish complex interactions that can either suppress or promote tumor progression [2]. Initially, the immune system attempts to eliminate malignant cells while minimizing tissue damage; however, in advanced disease stages, tumors exploit immune regulatory mechanisms to evade immune surveillance and promote malignancy [3].

Over the past 15 years, cancer immunotherapy has transformed clinical oncology, improved patient outcomes, and deepened our understanding of tumor biology. Immune checkpoint inhibitors (ICIs) and chimeric antigen receptor (CAR) T cell therapy have emerged as standard treatments across various cancer types and stages, highlighting the critical roles of the immune system in malignancies [4]. Despite their success, therapeutic resistance remains a major challenge, underscoring the need to explore the molecular mechanisms governing immune regulation [5].

Non-coding RNAs (ncRNAs) are increasingly being recognized as critical regulators of gene expression, exerting effects at the pre-transcriptional level—through epigenetic mechanisms and chromatin remodeling [6]—as well as post-transcriptionally by modulating RNA stability and protein interactions [7].

Based on their length, ncRNAs are categorized as small non-coding RNAs (<200 nucleotides), such as miRNAs and piRNAs, and long non-coding RNAs (lncRNAs, >200 nucleotides). Although lncRNAs lack protein-coding potential, they display structural similarities with mRNAs, being transcribed by RNA polymerase II, 5′-capped, and polyadenylated. These molecules are highly dynamic, localizing to various cellular compartments including the nucleus, cytoplasm, nucleolus, and mitochondria, where they participate in diverse regulatory functions, as detailed in Figure 1 [8].

Recent evidence has demonstrated that lncRNAs play a crucial role in regulating immune responses within the tumor microenvironment. Through modulating the expression of immune-related genes, lncRNAs influence key processes such as antigen presentation, cytokine signaling, immune cell activation, and checkpoint inhibition. Their ability to fine-tune immune evasion mechanisms makes them critical players in cancer progression and resistance to immunotherapy [9]. Notably, specific lncRNAs have been implicated in reshaping the balance between pro- and antitumor immunity, either by promoting an immunosuppressive environment or enhancing immune surveillance [10].

This review explores the diverse roles of lncRNAs in the cancer–immunity cycle, focusing on their impacts on immune cell activation, antigen presentation, and tumor–immune interactions. Understanding these regulatory mechanisms may pave the way for novel therapeutic strategies aimed at enhancing antitumor immunity and overcoming resistance to immunotherapy.

## 2. Cancer Immunosurveillance and Immunoediting

The relationship between inflammation and tumor cells has been recognized since the observations made by Virchow [11]. The immunosurveillance hypothesis was first proposed by Burnet and Thomas, who suggested that immune cells recognize and eliminate tumor cells based on abnormalities generated during carcinogenesis [12,13]. In particular, DNA alterations in tumor cells lead to the formation of neoantigens, which are absent in normal cells and trigger a specific immune response against these malignant cells [14].

Initially, the immunosurveillance hypothesis was viewed with skepticism due to the paucity of experimental evidence and subsequent observation that athymic CBA mice did not show an increased risk of developing cancer [15]. However, scientific and experimental advances—such as monoclonal antibody technology and various knockout mouse models involving various components of the immune system—have demonstrated the role of the immune system in tumor elimination. Experiments using mice deficient in immune components such as RAG2, type I/II IFN, Granzyme, or Perforin have revealed accelerated tumor growth, laying the foundations of cancer immunology and paving the way for immunotherapy [16].

Tumors can develop in individuals with intact immune systems, not only in those with aberrant immune system conditions. Dunn et al. introduced the concept of immunoediting, suggesting that innate and adaptive immune responses collaborate not only to recognize and eliminate tumor cells, but also to shape the immunogenicity of tumors [17]. Through the selective elimination of highly immunogenic tumor cells, the immune components impose selective pressure that allows less immunogenic or immunoevasive tumor cells to survive and proliferate, ultimately resulting in an immunoresistant tumor [18]. The efforts made and the knowledge gained from these studies have deepened our understanding of cancer-immune system relationships and have generated important contributions to the development of new clinical interventions.

### 2.1. Overview of the Cancer–Immunity Cycle and Immune Evasion

The cancer–immunity cycle describes the stepwise process through which the immune system recognizes and eliminates cancer cells. This cycle begins with the release of tumor antigens, which occurs when tumor cells undergo cell death due to intrinsic factors such as hypoxia and acidic pH, or are eliminated by cytolytic immune cells, including NK cells, NKT cells, macrophages, neutrophils, and γδ T cells—a specialized subset of intraepithelial lymphocytes [19].

Once released, tumor antigens are captured by local phagocytic cells such as immature dendritic cells (DCs) and macrophages. These cells degrade antigens into peptides within phagolysosomal vacuoles and then migrate to nearby lymph nodes, where they mature into antigen-presenting cells (APCs). In lymph nodes, APCs present tumor-derived peptides to CD4+ T cells, initiating their activation [20]. Simultaneously, through a process called cross-presentation, APCs activate CD8+ T cells, which acquire cytolytic functions through the influence of soluble cytokines [21].

Once activated, T cells travel to the tumor site, where they recognize and eliminate tumor cells expressing the corresponding antigens by employing their cytotoxic machinery. This destruction releases additional tumor antigens, perpetuating the cycle and sustaining the antitumor immune response [22]. However, continuous immune pressure selects tumor cells with reduced immunogenicity or those that acquire genetic alterations enabling immune evasion. Over time, this increased tumor heterogeneity disrupts the cancer–immunity cycle by impairing one or more of its steps, ultimately preventing an effective antitumor immune response [19].

The immune system has strict regulatory mechanisms that control autoimmune effector responses. However, the random and aberrant genetic transformation of malignant cells enables certain tumor clones to alter these regulatory pathways, allowing them to evade immune surveillance. Immune evasion is achieved through multiple strategies that promote a tolerogenic microenvironment [23].

The tumor undergoes reprogramming to meet increasing energy demands, which can directly affect the functionality of the surrounding immune cells. In parallel, there is a shift in cytokine production, with a reduction in pro-inflammatory cytokines and an increase in immunosuppressive cytokines such as interleukin (IL)-10, IL-6, and transforming growth factor-beta (TGF-β). This shift can result from either direct tumor cell activity or the reprogramming of infiltrating immune cells [24].

Moreover, tumors facilitate the infiltration and expansion of immunosuppressive regulatory cells, including regulatory T cells (Tregs) and myeloid-derived suppressor cells (MDSCs), while simultaneously inducing the polarization of immune cells toward pro-tumor phenotypes, such as Th2 cells, M2 macrophages, and N2 neutrophils [25].

Another key mechanism of tumor evasion involves the overexpression of co-inhibitory molecules, such as PD-1, CTLA-4, TIM-3, LAG-3, BTLA, and TIGIT, which drive T cell tolerance and lead to exhaustion or senescence [26]. Additionally, lymphocyte metabolism is affected by enzymes such as arginase-1, glutaminase, and indoleamine 2,3-dioxygenase (IDO), which further reduce their antitumor activity [27].

In advanced tumor stages, immune pressure selects tumor cells with reduced immunogenicity, either through downregulating antigen expression or altering antigen presentation pathways. These modifications include decreased expression of major histocompatibility complex (MHC) class I molecules or dysfunction of chaperone proteins such as TAP1, TAP2, and ERP57, ultimately impairing tumor recognition by the immune system [28].

### 2.2. Tumor Antigens: Release and Uptake

The cancer–immunity cycle begins with the release of tumor-specific or tumor-associated antigens (TAAs) from cancer cells. These antigens become accessible to the immune system either through active secretion or passive release during cell death. APCs—particularly dendritic cells—capture these TAAs and initiate T cell priming, ultimately leading to the recruitment and activation of effector cells that mediate tumor cell destruction [29]. Natural killer (NK) cells also play an essential role in immune surveillance, as they recognize and eliminate cells with altered or missing self-antigens—a hallmark of malignant transformation [30].

Among the different types of cell death, a specific subset known as immunogenic cell death (ICD) has garnered attention due to its ability to enhance antitumor immunity. ICD is a form of regulated cell death that, unlike non-immunogenic apoptosis, promotes the release of damage-associated molecular patterns (DAMPs) such as calreticulin (CRT), ATP, and high-mobility group box 1 (HMGB1), which act as adjuvants to stimulate dendritic cell activation and T cell responses [31,32].

Classical immunology postulates that the induction of an effective immune response requires APCs that phagocytose tumor antigens and simultaneously recognize them through receptor molecular signals released by cell death. These signals include pathogen-associated molecular patterns (PAMPs) and damage-associated molecular patterns (DAMPs) that arise from cell damage [33]. DAMPs are intracellular components that, when found in the membrane or released during ICD, are crucial for phagocytosis of dead cells by DCs; this, in turn, facilitates antigen presentation to T cells, thereby initiating an immune response against tumor antigens [34]. Examples of DAMPs are the “eat me” signals, such as the surface exposure of CRT, and other endoplasmic reticulum proteins, such as ERp57 [35].

Calreticulin (CRT) is an endoplasmic reticulum (ER) protein involved in various cellular functions including calcium homeostasis, protein folding, and MHC molecule assembly. Upon exposure to ICD inducers, CRT translocates from the ER lumen to the cell surface, becoming morphologically apparent before cell death. This translocation is dependent on the phosphorylation of eIF2α by protein kinase R (PKR)-like ER kinase (PERK) [36]. On the cell surface, CRT is an “eat me” signal for phagocytes, which recognize this signal via CD91 (also known as low-density lipoprotein receptor-related protein 1, LRP1) [37].

Research has indicated that CRT expression on the surface of acute myeloid leukemia cells stimulates the secretion of type I IFN by the CD8 + αDC subpopulation, which is necessary for T cell activation and induction of the immune response involved in tumor rejection [38]. CRT expression has been proposed as a prognostic marker in some types of cancer, as elevated expression correlates with better clinical outcomes. However, other studies have suggested a complex relationship, probably due to the intracellular roles of CRT in tumor progression and upregulation of the “don’t eat me” signal by the regulatory molecule CD47 [39].

As ICD is an antitumor response-promoting process, tumor cells often target ICD-associated DAMPs to disrupt the initiation of the cancer–immunity cycle. In many cancers, the overexpression of “don’t eat me” signals—such as those mediated by the CD47–SIRPα pathway—counteracts the pro-phagocytic signal of CRT, thereby preventing phagocytosis and hindering subsequent antigen uptake, ultimately disrupting the cancer–immunity cycle [40].

Additionally, “find-me” signals, such as ATP secretion, stimulatory signals for APCs, HMGB1 release, and type I interferon signaling, all contribute to the immunogenic nature of cell death [41]. However, the immunostimulatory effects of ATP can be neutralized through the overexpression of ectonucleotidases, such as CD39 and CD73, which degrade extracellular ATP into AMP and subsequently into immunosuppressive adenosine. Notably, CD73 is overexpressed in many tumor types and correlates with poor prognosis [42,43,44].

Tumor cells release antigens through various mechanisms, including passive shedding and vesicle formation, which occur independently of cell death [45]. In addition, several cancer therapies such as anthracyclines, oxaliplatin, radiation, and oncolytic viruses are known to induce ICD causing tumor cells to release tumor antigens and several DAMPs that stimulate immune responses [46,47,48,49].

### 2.3. Participation of LncRNA as a Regulator of DAMPs Expression

The following sections provide information on the role of lncRNAs in the regulation of some DAMPs.

#### 2.3.1. Calreticulin and CD47 Expression and lncRNAs: Regulating Phagocytosis

As mentioned above, phagocytosis of dying tumor cells is stimulated by “eat me” signals, in which CRT plays a preponderant role, whereas this process is inhibited by “don’t eat me” CD47 signals.

The non-coding RNA Retinoblastoma 1 (*ncRNA-RB1*), which is located upstream of RB1, plays a role in CRT regulation. Reports describe that knockdown of *ncRNA-RB1* in A549 lung carcinoma cells reduced CRT transcription without affecting RB1 protein expression. This reduction in CRT exposure diminished phagocytic uptake by CD11b+ macrophages in U2OS cells treated with mitoxantrone, a chemotherapeutic drug that induces ICD markers such as CRT exposure, ATP release, and HMGB1 release [50].

The Myocardial Infarction-Associated Transcript (*MIAT*), another lncRNA, has been involved in several pathologies [51]. *MIAT* has been implicated in the clearance of apoptotic cells in atherosclerosis by sponging *miR-149-5p*, which targets CD47 transcripts. Through sequestering *miR-149-5p*, *MIAT* increases the “don’t eat me” signals in CD47 expression and impairs apoptotic cell uptake [52]. *MIAT* is also overexpressed in breast cancer cell lines and biopsies, promoting cancer progression, whereas it is secreted in exosomes in the context of gastric cancer [53]. Thus, through modulating *ncRNA-RB1* and *MIAT* expression, tumor cells may impair antigen uptake by disrupting the balance between the pro- and anti-phagocytic signals provided by CRT and CD47.

#### 2.3.2. CD73 and lncRNAs: Immunosuppressive Mechanisms

CD73 is an ecto-5′-nucleotidase on the cell surface that breaks down AMP into adenosine, inhibiting T cell activation through the A2A receptor. CD73 is overexpressed in several types of cancers, promoting tumor survival and progression. CD73 knockdown restores antitumor T cell responses [54].

Small nuclear RNA host gene 16 (*SNHG16*) is overexpressed in multiple cancers—including hepatocellular carcinoma, bladder cancer, breast cancer, and osteosarcoma—and is correlated with tumor progression and poor survival outcomes [55]. In breast cancer, *SNHG16* is upregulated and secreted by exosomes. Upon reaching γδ T cells, *SNHG16* acts as a molecular sponge for *miR-16-5p*, which represses SMAD-5. This interaction upregulates TGF-β signaling and CD73 expression on γδ T cells, leading to immunosuppressive effects such as the inhibition of IFN-γ secretion in CD4+ T cells and reduced expression of perforin and granzyme B in CD8+ T cells, all mediated by the adenosine pathway [56]. Through the promotion of γδ CD73+ T cells, tumor cells hinder the activity of immune effector cells and the maintenance of the cancer–immunity cycle, as the tumor cells killed through ICD act as essential fuel for this cycle [57].

#### 2.3.3. HMGB1 and lncRNAs: Immune Activation and Evasion

HMGB1 is a chromatin-associated protein involved in DNA replication, repair, recombination, transcription, and genomic stability [41]. In ICD, HMGB1 is released as a danger signal, activating DCs via the TLR4/MyD88 signaling pathway and promoting the secretion of pro-inflammatory cytokines such as IL-1β, IL-6, TNF-α, and IL-8, which act as a “third signal” and are essential for initiating antitumor immune responses [58].

Although the overexpression of HMGB1 is important for inducing and maintaining inflammation, it also has a dual function. The absence of the lncRNA Semaphorin 3B (*SEMA3B*) antisense RNA 1 (*SEMA3B*-*AS1*) causes nucleocytoplasmic shuttling of HMGB1 for release into the extracellular space, increasing the production of cytokines IL-6 and IL-8, which are involved in promoting the proliferation, angiogenesis, invasion, and metastasis of gastric cancer [59]. This molecule has also been associated with cancer progression in hepatocellular carcinoma, lung cancer, breast cancer, and colorectal cancer [60]. Several lncRNAs—such as Metastasis-Associated Lung Adenocarcinoma Transcript 1 (*MALAT1*), Urothelial Cancer-Associated 1 (*UCA1*), HOX Transcript Antisense RNA (*HOTAIR*), and Zinc Finger E-Box Binding Homeobox 2 Antisense RNA 1 *(ZEB2-AS1*) —post-transcriptionally regulate the expression of HMGB1 [60]. For instance, *MALAT1* increases HMGB1 levels in hepatic injury, colorectal cancer, and multiple myeloma, often acting as a competitive endogenous RNA (ceRNA) for miRNAs that target HMGB1 [61]. *MALAT1* physically binds to HMGB1 and prevents its ubiquitination and degradation. Other lncRNAs, including *HOTAIR*, *ZEB2-AS1*, and *UCA1*, have been shown to regulate apoptosis and growth in various cancer types by sponging specific miRNAs that target HMGB1 expression [62].

#### 2.3.4. Immunogenic Cell Death (ICD), ER Stress Responses, and lncRNAs

As discussed above, the exposure of the cell surface to CRT is essential for the immunogenic properties of cell death. CRT induces the activation and maturation of APCs and the activation of the adaptive antitumor immune response; therefore, its detection has been proposed as a determining factor for the efficacy of ICD-inducing treatments. For instance, oxaliplatin—a well-studied ICD inducer—differs from cisplatin (CDDP), which does not induce ICD, in its ability to induce stress response. In this process, the phosphorylation of eukaryotic initiation factor 2alpha (*eIF2alpha*) is induced, triggering CRT translocation. Exposure of tumor cells to ER stressors, such as Thapsigargin, Tunicamycin, and Dihydroartemisinin, promoted CRT translocation even with CDDP treatment, restoring the immunogenicity of CDDP-induced cell death [63,64]. Interestingly, CRT exposure is dependent on endoplasmic reticulum (ER) stress, which involves the PERK/eIF2alpha pathway [36].

Unfolded protein response (UPR) is a homeostatic mechanism that is activated when ER stress occurs due to the accumulation of unfolded or misfolded proteins [65]. The interaction between ncRNAs and UPR components has been well documented, providing insights into the regulatory functions of lncRNAs under stress conditions. The LncRNA Upregulated in Colorectal Cancer (*LURCR*), which is highly expressed in colorectal cancer, promotes tumorigenesis through the regulation of UPR components [66].

ER stress-induced activation of PERK upregulates lncRNAs such as Golgin A2 pseudogene 10 (*GOLGA2P10*), which modulates Bcl-2 protein levels to protect hepatocellular carcinoma cells from apoptosis [67], and *MALAT1*, which enhances colorectal cancer cell migration following exposure to thapsigargin [68]. SOX2 is a master regulator of pluripotency genes. SOX2 overlapping transcript (*SOX2OT*) has been found to be overexpressed in breast cancer cells after exposure to ER stressors, such as thapsigargin or 4-hydroxytamoxifen, and downregulated UPR components such as BiP and PERK [69]. SOX2OT is linked to hepatocellular carcinoma invasion and osteosarcoma cell proliferation, and it is a potential prognostic marker for gastric cancer [70,71]. Its overexpression has also been reported in recurrent glioblastoma, where it promotes resistance to temozolomide—a chemotherapeutic agent known to induce CRT exposure and stimulate T cell activation in glioma models [72,73]. These lncRNAs are exploited by tumor cells to block the triggering of the immune response and their alteration represents a great opportunity to develop therapeutic strategies to improve the survival of cancer patients.

## 3. Cancer–Immunity Cycle and the Role of lncRNAs

Unlike most therapeutic strategies, which directly target tumor cells, immunotherapy indirectly exerts its effects through activation of the immune system. These treatments stimulate both innate and adaptive immune responses, creating a sustained challenge for tumors and significantly prolonging patient survival [74].

However, tumors in patients receiving immunotherapy can develop resistances, posing major challenges. Many patients fail to achieve durable clinical responses, underscoring the need to optimize or redesign innovative treatments. A deeper understanding of the mechanisms through which cancer cells evade antitumor immunity is crucial for the development of new therapeutic approaches. Enhancing the efficacy of immunotherapy could lead to better tumor control or, in some cases, complete eradication [75].

LncRNAs play crucial roles in antigen presentation, immune cell activation, effector function, and infiltration of immune cells within the TME [76,77], and are involved in multiple stages of the cancer–immunity cycle [78]. Thus, through influencing the recruitment and activation of innate and adaptive immune cells, lncRNAs reshape the tumor microenvironment, altering the immune landscape and promoting the tumor’s immune evasion strategies [78] (Figure 2).

One of the major contributions of lncRNAs to immune regulation is their role in the expression of immune-related genes, such as those relating to cell proliferation, epithelial–mesenchymal transition (EMT), and composition of the TME [79]. The tissue-specific expression and immunomodulatory capabilities of lncRNAs make them promising targets for cancer immunotherapy [80]. With advancements in single-cell technology, we now have deeper insights into cellular functions and regulatory pathways mediated by immune-related lncRNAs [81,82].

## 4. LncRNAs in Immune Modulation

The roles of lncRNAs in tumor development—particularly in terms of promoting or hindering the antitumor immune response—have recently gained attention. For instance, the lncRNA Long Intergenic Non-Coding RNA immunogenicity Trigger (LIMIT) has been identified as a regulator of tumor immunogenicity. LIMIT enhances immune recognition through activation of the GBP–HSF1 axis and increasing the expression of MHC class I molecules, thereby potentiating tumor immunogenicity and improving the efficacy of checkpoint therapy [83].

Additionally, lncRNA-derived peptides have been shown to enhance T-lymphocyte responses against tumors in mice [84]. Moreover, in conjunction with microRNAs (miRNAs), lncRNAs are emerging as therapeutic targets and biomarkers for the enhancement of cancer immunotherapy, offering promising avenues for the treatment of immune-refractory tumors [85]. These findings underscore the critical role of lncRNAs in cancer immunity and the associated potential for the development of novel immunotherapeutic strategies.

During antigen presentation, both immunogenic signals and functional APCs are required for the proper activation of T cells. The lncRNAs involved in the expression of various pro-inflammatory cytokines and their ability to regulate APC maturation are also indispensable. Research has shown that lncRNAs can significantly enrich the expression of MHC class I molecules and enhance antigen processing, suggesting that a diagnostic method based on lncRNAs and their alterations may be of great utility for the differential diagnosis of tumors and precancerous lesions [86].

Long Non-Coding RNA Dendritic Cell (*Lnc-DC*) is a specific marker located in the cytoplasm of the APCs [87]. Notably, Lnc-DC knockdown inhibited the activation of T lymphocytes [88]. Reports have indicated that *Lnc-DC* promotes the phosphorylation of the transcription factor STAT3 (signal transducer and activator of transcription 3) at tyrosine-705 by preventing STAT3 from being dephosphorylated by SHP1. Furthermore, *Lnc-DC* allows STAT3 to constantly translocate to the nucleus and maintain its transcriptional activity, confirming the regulatory effect of lncRNAs on antigen presentation [89,90].

Pro-inflammatory cytokines, such as interferon–gamma (IFN-γ) and tumor necrosis factor (TNF-α), act as regulators of antigen presentation [91,92]. Petermann et al. [93] have demonstrated that Th1 cells overexpress Interferon Gamma Antisense RNA 1 (*IFNG-AS1*) and synthesize IFN-γ. *IFNG-AS1* knockdown significantly decreased IFN-γ expression, whereas *IFN-γ* silencing did not affect *IFNG-AS1* expression. This suggests that IFN-γ expression is regulated by *IFNG-AS1* [93]. TNF-α induces DC maturation, and its formation is closely related to TNF-α and hnRNPL-related immunoregulatory lncRNA (*THRIL)*, which is expressed in macrophage-like cells. A feedback loop between *THRIL* and TNF-α has been shown, in which the silencing of THRIL decreases TNF-α expression, while high expression of TNF-α leads to the downregulation of *THRIL* [94].

As was previously mentioned, HMGB1 is released from dying cells induced by chemotherapy or radiotherapy. This molecule acts as a pro-inflammatory factor that induces DC maturation and antigen presentation [95,96]. Li et al. (2017) reported the existence of a relationship between HMGB1 and the Tumor Protein P73 Antisense RNA 1 (*TP73-AS1*) in hepatocellular carcinoma (HCC), which is required to stimulate the expression of HMGB1 and NF-κB pathway-regulated cytokines [97]. This group demonstrated that *TP73-AS1* and HMGB1 compete for binding to the *miR*-*200a* locus. Downregulation of *miR-200a* led to upregulation of HMGB1, RAGE, NF-κB, and pro-inflammatory cytokines. The authors reported that *miR-200a* was downregulated in HCC tissues, whereas *TP73-AS1* and HMGB1 were positively correlated, suggesting a possible therapeutic target [97].

Collectively, these lncRNAs—such as LIMIT, Lnc-DC, IFNG-AS1, and THRIL—modulate distinct yet complementary aspects of antigen presentation and dendritic cell maturation. While LIMIT enhances tumor immunogenicity via MHC class I upregulation, Lnc-DC and IFNG-AS1 act downstream by regulating key cytokines and transcription factors involved in T cell activation. Their activity converges on promoting effective immune priming, suggesting that their combined dysregulation may have synergistic effects in facilitating immune evasion. Understanding the temporal and cellular specificity of their action could inform more targeted immunotherapeutic strategies. Table 1 describes the inhibitory or promoter functions of lncRNAs during antigen presentation by APC and when triggering the activity of T lymphocytes involved in the cancer–immunity cycle.

### 4.1. Type I Interferon Signaling and lncRNA-Mediated Immune Regulation

DCs are central to initiating and sustaining antitumor immune responses. In the second step of the cancer–immunity cycle, DCs capture and process tumor antigens for presentation on MHC I and II molecules. DAMPs released by dead cells are critical for DC maturation, the expression of co-stimulatory molecules, migration to lymph nodes, and cytokine production, all of which are necessary for T cell activation against tumor antigens [100].

Tumor cells often express mechanisms that suppress the recruitment, maturation, and functioning of DCs, thereby hampering the cancer–immunity cycle. DC recruitment and subsequent T cell activation depend heavily on the production of chemokine CCL4 by tumor cells. High levels of CCL4 correlate with CD8+ T cell infiltration in tumors, which has been associated with better survival in patients with esophageal squamous cell carcinoma [101]. Consequently, exogenous CCL4 delivery recruits CD103 + DCs and CD8+ T cells, enhancing the efficacy of immune checkpoint blockade in multiple murine tumor models [102].

In a melanoma model, β-catenin activation in tumors led to reduced CCL4 expression through promotion of the transcriptional repressor ATF3, thereby limiting DC recruitment and diminishing responses to immune checkpoint inhibitors [103]. The lncRNA *mirR-4435*-2 host gene (*MIR4435-2HG*) strengthens β-catenin signaling by directly interacting with this protein and preventing its degradation, while also interacting with desmoplakin to inhibit the suppressive effect of β-catenin [104]. Importantly, it has been reported that MIR4435-2HG is overexpressed in lung, colorectal, hepatocellular, and gastric cancers, contributing to immune evasion [105].

Recent studies have highlighted the role of *HAND2-AS1* in regulating the tumor immune landscape. *HAND2-AS1* expression correlates with the infiltration of tumor-associated macrophages, NK cells, and T cells. Low expression of *HAND2-AS1* has been associated with an immunosuppressive tumor microenvironment, suggesting it may contribute to immune evasion and resistance to immunotherapy. Thus, modulation of *HAND2-AS1* could represent a promising strategy for enhancing immune infiltration and improving antitumor immunity [106].

Prostaglandin E2 (*PGE2*)—another suppressor of CCL4—inhibits DC recruitment and IL-12p40 production. In mice inoculated with COX2-deficient melanoma cells, tumor-infiltrating DCs showed enhanced expression of co-stimulatory molecules such as CD40 and CD86, along with elevated IL-12p40 production [107]. LncRNAs such as p50-Associated COX-2 Extragenic RNA (*PACER*), Highly Upregulated in Liver Cancer (*HULC*), *HOTAIR*, and MAF transcription factor G antisense RNA 1 (*MAFG-AS1*) are involved in regulating the expression of COX2 in various cancers, influencing immune suppression within the TME [108,109]. Multiple lncRNAs regulate COX-2 expression via different mechanisms. *PACER*, *HULC*, *HOTAIR*, and *MAFG-AS1* are lncRNAs that are broadly overexpressed in gastric, prostate, liver, and colorectal cancers and are correlated with poor prognosis [110,111]. For instance, *PACER* directly interacts with NF-κB p-50 and promotes its transcriptional activity by recruiting histone acetyltransferase p300 and RNApol II, which results in COX2 expression [112], while lncRNA *HULC* carcinoma epithelial in ovarian carcinoma [113], gastric cancer [114], cervical cancer [115], prostate cancer [116], and hepatocellular carcinoma [117] increase COX2 at the protein level by removing its polyubiquitin chains, diminishing its proteasomal degradation [118]. Instead, lncRNA *MAFG-AS1* and *HOTAIR* enhance COX2 expression by acting as ceRNAs that sponge *miR-101* and miRNA-143-3p, respectively, both of which target COX2 3′UTR and prevent cancer progression [119].

The DC phenotype is also influenced by lncRNA expression; for example, *MALAT1* induces a tolerogenic DC phenotype by interacting with NF-κB, inhibiting its binding to DNA, and functioning as a ceRNA for *miR-155*. This activity results in lower expression of CD80, CD86, MHC-II, IL-6, and IL-12 while enhancing production of the inhibitory cytokine IL-10 [120]. Conversely, the tolerogenic DC phenotype is induced by the downregulation of the LncRNA Nuclear paraspeckle assembly transcript 1 (*NEAT1*), which is characterized by low IL-1β production due to NLRP3 inhibition [121]. NEAT1 is overexpressed in lung, liver, breast, and colorectal cancers, but is downregulated in acute promyelocytic leukemia [122].

Furthermore, some lncRNAs encode micropeptides (see Section: 4.6) that have functional effects on DCs; for example, the lncRNA MIR155 host gene (*MIR155HG*)—which is highly expressed in DCs—encodes a 17-amino acid micropeptide that inhibits MHC-II antigen presentation by affecting HSC70 protein activity [123]. *MIR155HG* promotes macrophage polarization towards the M1 phenotype, whereas its inhibition promotes the M2 phenotype [124]. Moreover, overexpression of *MIR155HG* has been linked to the expression of the immune checkpoints PD-1, PD-L1, and CTLA-4 in several cancers [125].

Metabolically, DC dysfunction is associated with increased lipid storage. In cancer patients, it has been reported that DCs present a higher triglyceride content than in healthy individuals. The accumulation of oxidized lipids impedes antigen cross-presentation by diminishing MHC-I molecules on the DC surface [126]. Lipid accumulation is related to the higher expression of scavenger receptors such as MSR1. In this regard, through network analysis, it has been proposed that the lncRNA *cyp2c91* may contribute to MSR1 transcriptional regulation [127], whereas *MALAT1* induces the expression of scavenger receptor CD36 by activating β-catenin, which enhances lipid uptake in macrophages, thus attenuating DC maturation [128]. Moreover, MALAT*1* overexpression in models of atherosclerosis has also been shown to inhibit ox-LDL-stimulated DC maturation via the *miR-155-5p/NFIA* axis [129].

Type I IFN signaling is essential for the antitumor activity of DCs. IFNAR1 and STAT-1 knockout mice were unable to reject highly immunogenic tumors, and CD8α + DCs from these mice were defective in terms of cross-presenting to CD8+ T cells [130]. Therefore, the inhibition of type I IFN signaling enables cancer–immune evasion. Lung cancer-related transcript 1 (*LUCAT1*) limits the type I IFN response by directly interacting with STAT-1 in the cell nucleus and hindering the transcription of interferon-stimulated genes (ISGs) [131]. *LUCAT1* is overexpressed in gastric, esophageal, pancreatic, and lung cancers, and is associated with lower overall survival and advanced clinical stage [132]. Moreover, *LUCAT1* has been detected in exosomes in sera from lung adenocarcinoma (LUAD) patients [133].

In non-small cell carcinoma cells, a cluster of DCs co-expresses regulatory genes such as *CD274*, *PDCD1LG2*, and *CD200*, as well as maturity genes such as *CD40*, *CCR7*, and *IL*-*12b*. In this cluster of cells, tyrosine kinase receptor AXL signaling induces PD-L1 expression, similar to that in tumor cells [134]. Interestingly, AXL expression is regulated by lncRNAs, such as *MALAT1* in neuroblastoma [135]; DANCR is regulated by sponging *miR-33a-5p* in osteosarcoma [136]; and Glioma Stem Cell-Associated LncRNA (*GSEC*) is regulated by inhibiting *miR-202-5p* in triple-negative breast cancer [137].

Many lncRNAs have been reported to influence the recruitment, maturation, phenotype, and cytokine secretion ability of DCs, and the aberrant expression of some of these lncRNAs has been associated with cancer progression and pathogenesis. Further studies are needed to confirm the roles of lncRNAs in the inhibition of DC activity.

### 4.2. Micropeptides Encoded by lncRNAs: Emerging Players in Cancer Immunity

Long non-coding RNAs (lncRNAs) have traditionally been viewed as non-coding elements of the genome. However, recent advancements in sequencing technologies have revealed that many lncRNAs harbor short open reading frames (sORFs) that can encode functional micropeptides. These micropeptides have emerged as critical players in various biological processes, including immune regulation and cancer biology [138].

lncRNA-derived peptides can induce an immune response or have immunogenic properties. These micropeptides can be presented by class I major histocompatibility complex (MHC) molecules, making them potential targets for CD8+ T cells. Barczak et al. have shown that pharmacological inhibition of PRMT5 or modulation of E2F1 levels altered the repertoire of ncRNA-derived peptide antigens presented by tumor cells, thus improving their immunogenicity compared to untreated tumor cells. When presented to the immune system, these peptides can provoke a powerful response from antigen-specific CD8 T lymphocytes, causing significant delays in tumor growth [139]. Another example of how micropeptides encoded by ncRNA can serve as tumor-specific antigens is the study by Camarena et al., in which the authors analyzed RNA sequencing data from HCC tumors and identified that approximately 40% of the tumor-specific antigens are derived from non-canonical open reading frames (ncORFs) in ncRNAs. These peptides are shared by a significant proportion of HCC samples, making them promising candidates for cancer vaccines [140]

Certain micropeptides derived from ncRNAs have been shown to modulate antigen presentation pathways; for example, the micropeptide encoded by the lncRNA *MIR155HG* interacts with heat shock protein 70 (HSC70)—a chaperone involved in antigen trafficking and presentation in DCs. This interaction interrupts the HSC70–HSP90 machinery, thus modulating MHC class II-mediated antigen presentation and T cell priming [123]. Similarly, the Dleu2-17aa micropeptide, encoded by the Dleu2 ncRNA, promotes the generation of inducible regulatory T cells (iTreg) through improving the binding of Smad3 to region 1 of the conserved non-coding DNA sequence Foxp3 (CNS1), thus maintaining immunological homeostasis [141].

Micropetides derived from ncRNAs also play a role in innate immunity. It has been shown that a mitochondrial micropeptide, Mm47 (encoded by the ncRNA *1810058I24Rik*), is essential for activation of the Nlrp3 inflammasome. Cells deficient in Mm47 present altered responses from the Nlrp3 inflammasome, which highlights the importance of these micropeptides in the regulation of innate immunity [142].

Certain micropeptides derived from ncRNAs can counteract the immune evasion strategies employed by cancer cells; for example, the *LISRR* ncRNA suppresses the production of neoantigens while promoting the translation of PD-L1—a key player in immune checkpoint pathways. Inhibition of *LISRR* has been shown to stimulate antitumor immune responses, especially in melanomas resistant to immune checkpoint blockade (ICB) [143].

The role of micropeptides derived from ncRNAs in immunotherapy is an area of growing interest. These peptides can improve the efficacy of existing immunotherapies by modulating the functions of immune cells; for example, the micropeptide P155 (encoded by the ncRNA *MIR155HG*) has been shown to improve outcomes in mouse models of autoimmune inflammation via modulating antigen presentation [123].

## 5. T Cell Activation and lncRNA-Mediated Impaired T Cell Dysregulation

The complex nature of cancer–immune interactions is demonstrated by the great diversity of immune cells in the TME. While the cells involved in the innate immune response are key for the early elimination of tumor cells, the ultimate goal of the cancer–immunity cycle is the activation of the adaptive antitumor immune response. Most studies in this line have focused on the roles of CD4+ and CD8+ T cells and various T cell subsets (Th1, Th2, Th17, regulatory T cells (Tregs), T follicular helper cells (Tfh), and cytotoxic T lymphocytes) [144].

In the third step of the cancer–immunity cycle, DCs present tumor antigens via MHC class I and II molecules and produce cytokines, such as IL-12 and IL-1, to activate T cells. The activation of T cells mainly depends on three key signals provided by DCs in the microenvironment, which are mandatory for the correct activation of T cells. TCR recognition by tumor antigens represents the first signal, while TCR signaling is strengthened by the second signal provided by co-stimulatory molecules. CD28 is a prototypical co-stimulatory molecule expressed on T cells, which interacts with CD80 and CD86 expressed on DCs [145]. Cytokines at the activation site act as the third signal, stimulating T cell activation and ultimately defining the acquisition of their specific phenotype. In general, IL-2 is required for T cell proliferation, whereas other cytokines (e.g., IFN-γ, IL-4, TGF-β, IL-10) direct the differentiation of T cells towards distinct functional phenotypes, including Th1, Th2, Th17, and Tregs [145].

### 5.1. IL-2 Signaling and lncRNA Interactions

Interleukin-2 (IL-2) is a pleiotropic cytokine that mediates T cell proliferation, increases NK cytotoxic activity, induces the differentiation of regulatory T cells, and participates in activation-induced cell death (AICD). CD25 is the receptor for this cytokine, and the IL-2–receptor interaction stimulates downstream activation of the phosphoinositol 3-kinase (PI 3-K)/AKT, Ras–MAP kinase, and JAK–STAT pathways, which are involved in cell proliferation, survival, AICD, and differentiation. IL-2 modulates the expression of its receptor and receptors for other cytokines and transcription factors, thereby promoting CD4+ and CD8+ T cell differentiation [146].

In bone marrow-derived mesenchymal stem cells derived from patients with systemic lupus erythematosus, an interaction between the lncRNA H19 and the IL-2 transcript has been reported. Overexpression of H19 reduced IL-2 mRNA and protein levels via base-pairing with its transcript [147]. Overexpression of H19 has been reported in multiple cancers, which promotes tumorigenesis by promoting proliferation, stress survival mechanisms, EMT, and metastasis [147]. Furthermore, H19 can be secreted into exosomes, which has been reported to promote the chemoresistance of tumor cells against oxaliplatin and doxorubicin, both in vivo and in vitro. Interestingly, in a mouse colorectal model, it has been shown that exosomal H19 is not produced by tumor cells but, instead, by cancer-associated fibroblasts (CAFs) [148]. Thus, overexpression of the lncRNA H19 in the tumor microenvironment may impair T cell proliferation through diminishing IL-2 levels.

Bioinformatic studies have suggested that the lncRNA OPA-interacting protein 5 antisense transcript 1 (*OIP5-AS1*) controls the expression of CD25, the alpha chain of the IL-2 receptor, and its downstream molecules by sponging multiple miRNAs [149]. *OIP5-AS1* dysregulation has been associated with tumor cell proliferation, survival, invasion, migration, EMT, and metastasis, although its prognostic value remains controversial [150]. For instance, upregulation of *OIP5-AS1* has been reported in multiple cancers, such as glioma, hepatoblastoma, and osteosarcoma, whereas multiple myeloma and radio-resistant colorectal cancers present decreased expression of this lncRNA [151]. Significantly, OIP5-AS1 can be secreted into exosomes by tumor cells and CAFs, and has been reported to promote cancer progression, angiogenesis, and autophagy [152,153].

Together, these findings suggest that tumor- and stromal-derived lncRNAs may disrupt IL-2-mediated immune activation at multiple levels—both by reducing IL-2 production and by impairing IL-2 receptor signaling—thereby contributing to immune evasion within the tumor microenvironment.

### 5.2. CTLA-4 Regulation by lncRNAs

The lack of a second signal frequently results in impaired activation, anergy, or even apoptosis of T cells. The immune checkpoint CTLA-4 is a critical co-inhibitory molecule that blocks T cell activation through competing with the co-stimulatory signal mediated by CD28 for binding to CD80 and CD86 expressed in DCs. In CD4+ T cells isolated from patients with asthma, a correlation between CTLA-4 expression and lncRNA *MALAT1* has been reported. As a ceRNA, *MALAT1* sequesters *miR-155*, leading to increased CTLA-4 expression and the promotion of a Th2-biased immune response. Given that *MALAT1* is overexpressed in various types of cancer and is secreted in exosomes, it is tempting to speculate that its overexpression prevents the activation of antitumor T cells [154].

### 5.3. GAS5: Tumor-Suppressor lncRNA in Immune Regulation

Growth Arrest-Specific 5 (*GAS5*) codes for an lncRNA, as well as some snoRNAs, miRNAs, and piRNAs. The *GAS5* lncRNA is acknowledged to be a tumor-suppressor gene (TSG) that hinders glucocorticoid receptor signaling by sponging *miR-21* and regulating translation [155]. A diminished expression of *GAS5* compared to that in normal tissue has been reported in multiple cancers, and this downregulation has been related to diminished apoptosis, increased proliferation, and poor prognosis in breast, pancreatic, cervical, bladder, and gastric cancers, as well as NSCLC [156]. Additionally, tumor cells may promote *GAS5* downregulation to evade immune elimination, as knockdown of *GAS5* impairs IFN-γ and TNF-α secretion and the cytotoxic activity of NK cells against hepatocellular carcinoma and MGC-803 cells, while its overexpression favors tumor rejection in a xenograft mouse model [157]. *GAS5* acts by sponging *miR-544*, which targets the RUNX3 transcription factor that is essential for NK cell functions [158], as well as *miR-18a*, which has been previously reported to inhibit NK cytotoxicity by downregulating the activating receptor NKG2D [159].

Regarding T cell activity, *GAS5* presents a more complex role. The expression of *GAS5* induced growth arrest in normal lymphocytes stimulated with phytohemagglutinin [160], and the inhibitory effect of rapamycin and other paralogues on T cell proliferation has been shown to largely depend on *GAS5* expression [161]. In systemic lupus erythematosus, it has been reported that *GAS5* inhibits CD4+ T cell activation by sponging *miR-92a-3p*, which targets E4 binding protein 4 (E4BP4) [162]. Moreover, co-incubation of exosomal *GAS5* with CD4+ T cells inhibited their differentiation into an IFN-γ-producing Th1 phenotype, skewing it towards an IL-4-producing Th2 phenotype, in part by downregulating the transcription factor T-bet, which may promote tumor cell survival [163]. Intriguingly, *GAS5* has been reported to be upregulated in exosomes derived from colorectal cancer patients, although it has been reported to be downregulated in exosomes derived from the sera of NSCLC patients and in tumor mouse models [164].

Conversely, *GAS5* acts by sponging *miR-21* [165], which has been reported to negatively regulate T cell activation [166], suggesting that *GAS5* downregulation in the tumor microenvironment may promote *miR-21* activity and, therefore, impair T cell activation. Diminished expression of *GAS5* and augmented expression of *miR-21* have been reported in T cells from HIV patients. Moreover, overexpression of *GAS5* increases IL-2- and IFN-γ-producing CD4+ T cells [166].

### 5.4. LncRNA-Mediated Regulation of T Cell Phenotypes

The cytokine environment at the activation site plays a critical role in determining the phenotypes of T cells. IL-12, IL-2, and IFN-γ promote Th1 differentiation, which supports antitumor immunity, whereas IL-4, IL-5, and IL-10 drive Th2 differentiation, which is associated with tumor growth and immune suppression. Cytokines such as Il-6 and TGF-β induce differentiation to Th17, which is associated with chronic inflammatory conditions, whereas anti-inflammatory cytokines IL-10 and TGF-β induce Tregs differentiation, which suppresses immune responses and aids in tumor evasion [167].

Whole-genome RNA sequencing has identified specific lncRNAs that influence T cell polarization under Th1, Th2, or Th17 conditions. These lncRNAs are often co-expressed with nearby phenotype-specific, protein-coding genes. For example, lncRNA T Cell Master Enhancer Viral Integration Site 1 (*Tmevpg1*) has been shown to regulate Th1 differentiation, while GATA Binding Protein 3 Antisense RNA 1 (*GATA3-AS1*) promoted Th2 differentiation [168]. Tumor cells can exploit these lncRNAs to favor pro-tumor Th2 and Tregs phenotypes, impairing the differentiation of naïve T cells into antitumor Th1 cells.

*GATA3*—a master regulator of Th2 differentiation—promotes IL-4, IL-5, and IL-13 production. The lncRNA *GATA3-AS1*, which is encoded adjacent to GATA3, facilitates an open chromatin state at the *GATA3* locus and enhances Th2-associated cytokine expression. *GATA3-AS1* overexpression has been reported in breast, pancreatic, and hepatocellular cancers and is linked to tumor progression and metastasis. Importantly, in triple-negative breast cancer, *GATA3-AS1* has been suggested to promote immune evasion by inducing the de-ubiquitination of PD-L1, a critical immune checkpoint [169]. Similarly, Guanine Nucleotide-Binding Protein Alpha Stimulating Antisense RNA 1 *(GNAS-AS1)*, which is overexpressed in estrogen receptor-positive breast cancer cells, promotes M2 macrophage polarization by sponging *miR-433-3p*, which targets *GATA3*. This relationship suggests that lncRNAs regulating *GATA3* may facilitate immune escape mechanisms through the promotion of Th2 and M2 phenotypes in the TME [170].

Induction of Tregs, which are essential for maintaining immune homeostasis, is another critical immune escape strategy that is widely used by tumors. Tregs impair antitumor immunity by competing for IL-2, inhibiting APCs through CTLA-4 expression, producing immunosuppressive cytokines, degrading ATP to adenosine, and eliminating effector cells via granzyme release. The expression of FOXP3—the master regulator of Tregs differentiation—is regulated by lncRNAs such as FOXP3 Long Intergenic Non-Coding RNA (*Flicr*), which modulates FOXP3 expression by influencing chromatin accessibility [171].

A correlation between FOXP3 and the lncRNA *DQ786243* on CD4+ T cells has been reported in chronic inflammatory diseases, such as Crohn’s disease. Transfection of Jurkat cells with the lncRNA *DQ786243* increased mRNA and protein levels of the FOXP3 transcription factor, likely through the promotion of cAMP response element binding protein (CREB) activation via phosphorylation [172]. Mechanistically, *DQ786243* acts as a ceRNA of *miR-506*, which targets CREB by directly binding to its 3′-UTR. In ovarian and gastric cancers, overexpression of *DQ786243* has been related to poor clinical prognosis, while its inhibition in ovarian cancer cells suppressed proliferation, migration, and invasion [173].

Other lncRNAs have also been reported to regulate Tregs differentiation; for instance, the oncogenic lncRNA *HULC*—which has been associated with poor overall survival and metastasis in multiple tumors [174]—has been reported to promote Tregs differentiation when transfected into T cells from healthy donors in vitro [175].

Downregulation of the lncRNA Maternally Expressed Gene 3 (*MEG3*) in T cells from asthmatic and autoimmune thrombocytopenic purpura patients inhibited Th17 cell differentiation. *MEG3* functions as a ceRNA, sequestering miR-17 and miR-125a-5p, both of which target RORyt mRNA. This regulation leads to increased RORyt expression and promotes FOXP3 upregulation, thereby influencing T cell differentiation [176]. Notably, MEG3 has been reported to be downregulated in gastric and esophageal cancers, likely due to aberrant hypermethylation of its promoter [177]. This downregulation may serve as a tumor-driven mechanism to facilitate immune evasion through the promotion of Tregs differentiation, ultimately contributing to an immunosuppressive TME.

In contrast, FOXF1-Adjacent Non-Coding Developmental Regulatory RNA (*FENDRR*) has been reported as a TSG that prevents tumor progression in colon cancer and NSCLC [178] and sensitized osteosarcoma cells to doxorubicin [179]. *FENDRR* acts as a ceRNA that sponges *miR-423-5p*—a miRNA that targets explicitly GADD45b—preventing the secretion of TGF-β, VEGF, and IL-10 in HCC cells and the immunosuppressive activity of Tregs [180]. For the design of novel therapeutic strategies, it is essential to further investigate and deepen the understanding of these and other lncRNAs in the induction of different T cell sub-populations.

*HOTTIP* is another lncRNA implicated in immune escape mechanisms. It promotes the expression of IL-6, which not only facilitates tumor proliferation and survival but also modulates immune evasion. Elevated IL-6 levels upregulate PD-L1 expression on neutrophils, thereby contributing to the inhibition of T cell proliferation and activation [181]. This pathway exemplifies how tumor-derived lncRNAs can reprogram innate immune cells to suppress adaptive immune responses [182].

Despite acting through different mechanisms—such as chromatin remodeling (*GATA3-AS1*), miRNA sponging (*DQ786243*, *HULC*, *MEG3*), or checkpoint regulation (*HOTTIP*)—many of the lncRNAs described in this section converge functionally to promote Th2 and Treg differentiation. This convergence supports tumor-induced immune tolerance and suppression of effective antitumor Th1 responses. Conversely, FENDRR and Tmevpg1 serve as counter-regulatory elements that favor inflammatory phenotypes. The opposing roles of these lncRNAs highlight potential redundancies and antagonisms that could be exploited to reprogram T cell responses within the tumor microenvironment.

## 6. LncRNAs in T Cell Trafficking

The fourth and fifth steps of the cancer–immunity cycle involve the trafficking of activated T cells to the tumor, their extravasation from blood vessels, and subsequent infiltration into the tumor parenchyma [183]. This process is regulated in the TME through the production and release of growth factors and cytokines by fibroblasts, myeloid cells, endothelial cells, and tumor cells, which collectively influence immune cell infiltration. Several chemokines promote transendothelial migration and enhance T cell infiltration into tumors, with their expression often correlating with better clinical outcomes [183].

In contrast, endothelial cell anergy has emerged as a tumor-induced mechanism that restricts T cell infiltration. Pro-inflammatory cytokines such as TNF-α and IL-1β upregulate adhesion molecules, including E-selectin, ICAM1, and VCAM1, and induce chemokine secretion by endothelial cells, thus facilitating immune cell extravasation. However, in the context of endothelial anergy, prolonged exposure to angiogenic factors—particularly VEGF—inhibits these pro-inflammatory effects, leading to reduced adhesion molecule expression and impaired NK and T cell infiltration. This phenomenon has been proposed as a vascular immune checkpoint, co-opted by tumors from mechanisms originally involved in immune privilege during embryonic development [184,185]. Antiangiogenic agents that target the VEGF–VEGFR pathway have shown promising results in reversing tumor-induced endothelial anergy, thereby restoring T cell infiltration and immune-mediated tumor destruction.

Multiple lncRNAs regulate VEGF expression in tumor cells under various conditions, including Testis Development-Related Gene 1 (*TDRG1*), *MEG3*, *H19*, *SNHG1*, Non-Coding RNA Activated by DNA Damage (*NORAD*), Antisense Non-Coding RNA in the INK4 locus (*ANRIL*), and *MALAT1*. Dysregulation of these lncRNAs may contribute to tumor-induced endothelial anergy through the modulation of VEGF expression, further restricting immune cell infiltration into tumors [186,187].

Overexpression of the lncRNA *TDRG1* has been reported to promote the proliferation, migration, invasion, and metastasis of cervical carcinoma [188]; promote gastric cancer through modulation of the expression of hepatoma-derived growth factor (HDGF) [189]; and promote NSCLC with involvement of the zinc finger e-box binding homeobox 1 (*ZEB1*) [190]. In human retinal microvascular endothelial cells, hyperglycemia induced VEGF expression, while knockdown of *TDRG1* reversed this effect [191]. Furthermore, in endometrial adenocarcinoma cells, it has been shown that *TDRG1* directly interacts with VEGF-A protein and that its overexpression augments the protein levels of VEGF-A, whereas knockdown of this lncRNA diminished VEGF-A protein [192]. Conversely, in normal endothelial cells, it has been reported that high glucose concentrations induce VEGF expression, while the lncRNA *MEG3* inhibits this effect [193]. As pointed out earlier, the expression of *MEG3* is downregulated in multiple cancers; therefore, tumor cells may promote VEGF expression through downregulation of this lncRNA.

Collectively, these findings highlight the contrasting roles of TDRG1 and MEG3 in the regulation of VEGF-A expression. While TDRG1 promotes tumor progression and immune evasion by upregulating VEGF-A levels, thereby potentially contributing to endothelial anergy and impaired immune cell infiltration, MEG3 acts as a tumor suppressor by repressing VEGF expression under stress conditions such as hyperglycemia. In the context of cancer, the frequent downregulation of MEG3 and overexpression of TDRG1 may synergistically enhance VEGF-driven vascular remodeling, facilitating immune escape and tumor dissemination. Thus, targeting VEGF-regulating lncRNAs may offer a promising strategy to restore immune infiltration and improve antitumor immunity. These functional roles are further summarized in Table 2.

## 7. Metabolic Reprogramming and lncRNA-Mediated Immune Escape

Metabolic reprogramming plays a central role in tumor progression, not only by supplying the energy and biosynthetic precursors necessary for uncontrolled proliferation, but also through alteration of the TME in ways that impair antitumor immunity. One hallmark of this reprogramming is the Warburg effect, in which cancer cells preferentially utilize glycolysis even in the presence of oxygen. This metabolic shift supports rapid growth and leads to the accumulation of immunosuppressive metabolites such as lactate, which inhibit the function of infiltrating immune cells [197].

Beyond glycolysis, metabolic alterations in cancer cells result in the production of immunomodulatory compounds and can reduce the expression of MHC class I molecules, thereby hindering antigen presentation and impairing T cell recognition [198]. The TME becomes enriched in metabolites such as lactate and kynurenine, further suppressing immune responses and contributing to immune evasion [198].

Recent studies have underscored the intricate interplay between cancer metabolism, immune suppression, and ncRNAs, particularly lncRNAs. LncRNAs are key regulators of cancer cell metabolism, influencing the activities of enzymes and signaling pathways involved in glycolysis, fatty acid metabolism, and oxidative phosphorylation [199]. In addition to shaping metabolic programs, lncRNAs modulate immune cell differentiation and function, thereby influencing immune surveillance and evasion mechanisms within the TME [200].

The TME itself is a complex ecosystem composed of tumor cells, immune cells, stromal cells, and endothelial cells, which interact through a dynamic network of cytokines, growth factors, and metabolites [200]. Tumor cells promote chronic antigenic stimulation that may either activate or suppress immune responses, depending on the cellular and molecular context [201]. Furthermore, tumor-derived signals released during immunogenic or necrotic cell death can modulate immune responses and promote tumor tolerance [202].

Importantly, high infiltration of memory Th1 and cytotoxic CD8+ T cells has been associated with better clinical outcomes, emphasizing the potential of immune-mediated tumor control [203]. In this regard, “tumor microenvironment antigens (TMAs)” have emerged as promising immunotherapeutic targets, offering potential advantages over conventional tumor-associated antigens [204]. Moreover, tumor cells can release microvesicles that transfer oncogenic material and modulate the phenotype of recipient cells, further contributing to immune evasion and tumor heterogeneity [205].

Immunosuppressive mechanisms in the TME are often reinforced by regulatory cell populations such as Tregs, MDSCs, and cancer-associated fibroblasts, which work in concert to inhibit effective antitumor immunity [206]. Hypoxia and other metabolic stressors in the TME drive additional immunosuppressive pathways and promote tumor survival [207]. Targeting metabolic and immunological alterations mediated by lncRNAs offers new therapeutic possibilities, especially those aimed at reprogramming EMT and improving the efficacy of antitumor immune responses.

## 8. Extracellular Vesicles and Intercellular Communication

Intercellular communication is composed of ligand–receptor (autocrine, paracrine, and endocrine) and cell–cell (juxtacrine) signaling mechanisms. It has recently been described that cells exchange information by releasing membrane-enclosed particles called extracellular vehicles (EVs) [208]. EVs contain proteins, nucleic acids, lipids, and soluble and insoluble signaling factors. In addition, they can travel through body fluids to distant sites.

As described above, lncRNAs such as *MIAT, SNHH16, LUCAT1, OIP-AS1, MALAT-1, GAS5,* and *H19*, which regulate functions that might disrupt different steps of the cancer–immunity cycle, have been found in EVs. Therefore, by secreting EVs, tumor cells might employ these lncRNAs to hinder the antitumor immune response in the TME and in distant sites, such as draining lymph nodes or preparing the pre-metastatic niche.

Moreover, EVs have been reported to promote chemoresistance in hepatocellular carcinoma. Under chemotherapeutic stress, high expression of Long Intergenic Non-Coding RNA-Regulator of Reprogramming (*LINC-ROR*) in EVs promotes the proliferation of CD133+ cells by TGF-β and consequently weakens the effect of chemotherapy drugs [209]. In contrast, EVs associated with Long Intergenic Non-Coding RNA–Very Low-Density Lipoprotein Receptor (*LINC-VLDLR*) can enhance the efficacy of chemotherapy through the regulation of intercellular communication. VLDLR (Very Low-Density Lipoprotein Receptor) has been found to enhance the expression of ATP-binding cassette sub-family G member 2 (ABCG2) by *LINC-VLDLR*, decrease cell cycle progression, reduce cell viability, and promote chemotherapy-related death [210].

This dual role of EVs underscores their complexity as mediators of cancer progression and therapeutic resistance, suggesting their utility as potential targets or tools for innovative therapeutic strategies. Understanding the molecular mechanisms underlying EV-mediated intercellular communication will be pivotal for harnessing their full potential in combating chemoresistance and improving therapeutic outcomes.

## 9. Therapeutic Potential of lncRNAs in Cancer Immunotherapy

Given the diverse regulatory roles associated with the immune response, lncRNAs are emerging as potential targets for the improvement of cancer immunotherapy. Through the modulation of gene expression, immune cell differentiation, and the TME, lncRNAs influence several steps in the cancer–immunity cycle, making them attractive candidates for therapeutic intervention.

One promising approach involves targeting lncRNAs that regulate immune checkpoint expression, such as PD-L1 and CTLA-4, in order to enhance the effectiveness of checkpoint inhibitors. For instance, the inhibition of these lncRNAs could restore T cell functionality and improve responses to immune checkpoint blockade therapies. Another promising strategy is the reversal of immune suppression. LncRNAs have been reported to support immunosuppressive phenotypes in Tregs and M2 macrophages, facilitating tumor immune evasion; for example, *GNAS-AS1* promotes M2 macrophage polarization by sponging of miR-433-3p [211], while *DQ786243* is linked to FOXP3 expression in Tregs [172]. Inhibiting these lncRNAs could limit the presence of immunosuppressive cell populations within the TME, fostering a more effective antitumor immune response.

LncRNAs also regulate ICD, a key mechanism that triggers the antitumor immune response. LncRNAs such as *MIAT* and *ncRNA-RB1* modulate phagocytic signals such as CRT and CD47, which are critical for ICD. Targeting these lncRNAs may enhance antigen release and improve the efficacy of ICD-inducing therapies, including radiotherapy and chemotherapy. Additionally, lncRNAs represent potential biomarkers or therapeutic targets to enhance existing treatments; for example, the exosomal lncRNA *H19*—which is secreted by cancer-associated fibroblasts—has been associated with chemoresistance. Inhibiting H19 could not only improve the sensitivity to chemotherapy, but may also enhance immune responses when combined with immune checkpoint inhibitors [212].

The rise of single-cell technologies and bioinformatics has significantly improved the identification of immune-related lncRNAs, paving the way for the development of personalized immunotherapy. Future research should focus on integrating lncRNA profiling with immune checkpoint inhibition and TME remodeling to optimize clinical outcomes. Despite these promising insights, challenges remain, particularly regarding the specificity of lncRNA-targeting therapies and potential off-target effects [213]. Further studies integrating lncRNA profiling with existing immunotherapeutic strategies will be crucial for translating these discoveries into clinical applications. However, the clinical translation of lncRNA-targeted therapies also faces delivery challenges, including immune recognition, rapid clearance, and limited tissue specificity—especially given the size and complexity of many lncRNAs [214]. Novel chemistries and delivery systems are under development, but overcoming these barriers remains critical [215].

## 10. Concluding Remarks

LncRNAs are a class of RNAs that regulate diverse biological functions at the epigenetic, transcriptional, and pre- and post-transcriptional levels. In the cancer–immunity relationship, lncRNAs may favor the antitumor immune response or promote tumorigenesis and participate in tumor-mediated evasion mechanisms. Their roles seem to depend mainly on the stage of tumor development, the type and grade of various tumor-infiltrating immune cells, and intratumoral heterogeneity, among other variables. Currently, the crucial role of lncRNAs has been analyzed in tissues, serum, plasma, and other biological components using various technologies such as qRT-PCR, microarrays and RNAseq. However, with the rise of single-cell technologies and bioinformatics tools, substantial advancements have been made in understanding the role of lncRNAs in the relationship between the immune system and cancer. Although progress has been made in understanding the implicated relationship of lncRNAs in cells and molecules involved in several signaling pathways in the tumor microenvironment, there is a need to better understand the intricate regulatory function of lncRNAs and their association with other genomic alterations, with the goal of inducing a potential response in immune–tumor communication. Future research on lncRNA-targeted therapies may pave the way for personalized immunotherapeutic approaches, in which treatments are tailored based on each patient’s unique lncRNA expression profile. These approaches could significantly improve clinical outcomes by combining immune checkpoint inhibition that triggers a targeted immune response against the tumor with strategies to modulate lncRNAs involved in immune escape.

## Figures and Tables

**Figure 1 ijms-26-04821-f001:**
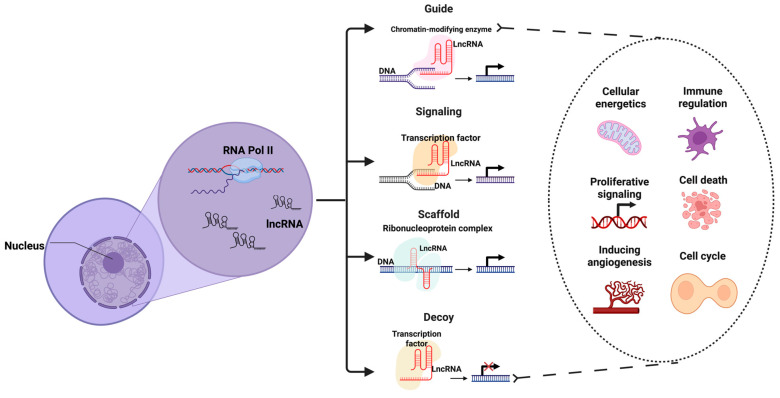
Principal functions of lncRNAs in the cell. Black arrows indicate activation or promotion of gene transcription, whereas red crosses represent transcriptional repression or inhibition. Created using https://BioRender.com (accessed on 16 May 2025).

**Figure 2 ijms-26-04821-f002:**
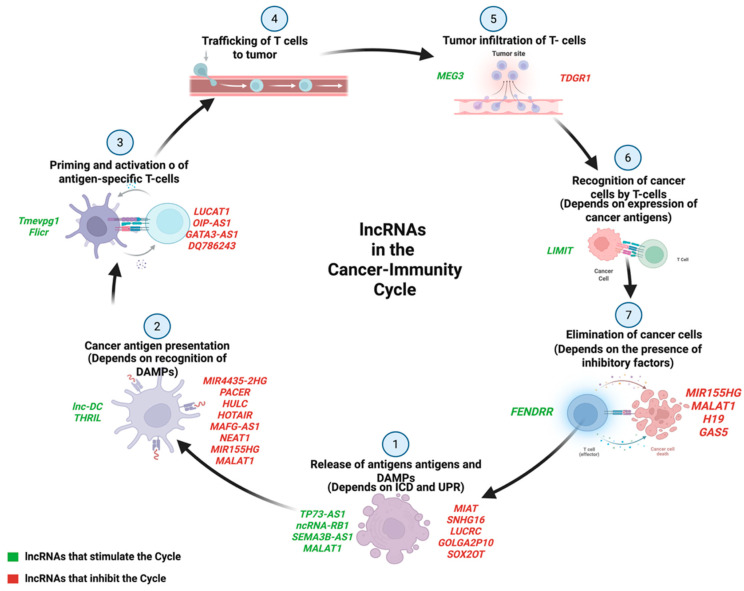
Integration of long non-coding RNAs (lncRNAs) in the cancer–immunity cycle. LncRNAs participate in multiple steps of the cancer–immunity cycle, either promoting (green) or inhibiting (red) antitumor immunity. These include antigen release and presentation, as well as T cell activation, trafficking, infiltration, and cytotoxicity. Created using https://BioRender.com (accessed on 16 May 2025).

**Table 1 ijms-26-04821-t001:** Regulatory roles of lncRNAs in immune cell-mediated functional activities.

LncRNA	Effect on Immune Cells	Reference
*MIAT*	Overexpresses CD47, blocking macrophage-mediated phagocytosis	[52]
*SNHG16*	Inhibits CD4 and CD8 T cell activity	[56]
*SEMA3B-AS1*	Reduces inflammatory cytokines	[59]
*ncRNA-RB1*	Overexpresses CRT, promoting antigen presentation	[85]
*Lnc-DC*	Stimulates DC differentiation and T cell activation	[90]
*TP73-AS1*	Overexpresses HMGB1	[97]
*IFNG-AS1*	Increases IFN-γ expression	[98]
*lnc-THRIL*	Regulates TNF-α secretion	[99]

**Table 2 ijms-26-04821-t002:** Summary of the effects of lncRNAs in the tumor microenvironment (TME). Reports indicate that in immune cells, they induce opposing activities, promoting or hindering cell activation, while in the tumor, they promote tumor progression by facilitating tumor escape from the host immune response.

	Effect on Immune Cells	Effect on Cancer Cells	Reference
*LURCR*	-	Promotes tumorigenesis.	[66]
*GOLGA2P10*	-	Inhibits cell death.	[67]
*SOX2OT*	-	Induces tumor resistance.	[69]
*PACER, HULC,* *HOTAIR, MAFG-AS1*	Regulate COX-2 expression.	-	[109,111,113,119]
*HULC, lnc-EGFR*	Promote Tregs differentiation.	-	[113,194]
*HOTAIRM1, ZEB2-AS1*	Reduce DC differentiation.	Promote proliferation, progression, migration, and apoptosis inhibition of cancer cells.	[119,150]
*NEAT1*	Increases inflammatory cytokines and co-stimulatory molecules.	Correlated with proliferation, metastasis, survival, and drug resistance.	[122]
*MIR155HG*	Promotes antigen presentation.	Increases PD-1, PD-L1, and CTLA-4 in hypoxic conditions.	[123]
*MALAT1*	Increases HMGB1 and reduces inflammatory cytokines.	Induces tolerogenic dendritic cells and promotes tumor growth.	[128,129]
*LUCAT1*	Reduces Type I interferon.	Promotes tumor progression.	[131]
*H19*	Reduces IL-2 production.	Promotes resistance in cancer cells.	[147]
*OIP5-AS1*	Reduces CD25 expression.	Promotes cancer development and progression.	[150]
*GATA3-AS1*	Favors Th2 differentiation.	-	[169]
*DQ786243*	Induces Tregs by stimulating FOXP3 expression.	-	[172]
*FENDRR*	Promotes pro-inflammatory cytokine production.	-	[178]
*MEG3*	Promotes Tregs differentiation, blocks Th17 cell differentiation.	-	[193]
*MIR4435-2HG*	-	Facilitates immune evasion.	[195]
*Tmevpg1*	Favors Th1 differentiation.	-	[196]

## Data Availability

Not applicable.

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
