# Peer review of "A Comprehensive Review of Long Non-Coding RNAs in the Cancer–Immunity Cycle: Mechanisms and Therapeutic Implications"

_ijms, 2025, doi:10.3390/ijms26104821_

Round 1

Reviewer 1 Report

Comments and Suggestions for Authors

Kindly see the attached comments' file

Comments on the Quality of English Language

A thorough English editing is required, please see more details in the attached.  

Author Response

Response to Reviewer 1, Manuscript IJMS 3513854

The authors are very grateful for the comments made on the manuscript. The following is a list of the changes made following the sequence sent to us:

Major comments:

  1. 1 line 40. Non-coding RNAs (ncRNAs have emerged as key post-transcriptional regulators of gene expression. It is not only post-transcriptional but can also be pretranscriptional

In this new version, Line 43-44. The pre-transcriptional role of ncRNAs is included.

  1. Lines 142-148. It seens like ICD is not properly defined and is only mentioned and used in text as part of a broader discussion.

According to your suggestion, the definition of the immunogenic cell death (ICD) and some molecules which acts as adjuvant to stimulate the adaptive immune response are indicated in the paragraph, lines  158-163.

  1. Lines 154-156. DAMPs, the word serial is not a good description of DAMPs. The text suggested was included in this version. Other forms of cell death also release DAMPs.

In lines 168-171, the information was restructured according to your proposal and other agents that induce ICD are indicated in lines 190- 197.

To clarify the information and make it more comprehensive to read, we decided to change the sequence initially proposed.  Now, we indicate information on the balance of phagocytosis mediated by CRT as a positive regulator and CD47 as a negative regulator. Lines 198- 232. 

  1. Lines 170-172. It is not just tumor cells but also tumor microenvironment cells that contribute to immune evasion. Please amend to clarify.

Information respect to evasion mechanisms of tumors in which lncRNAs are implicated is indicated along the text. This information is contrasted to the antitumoral activity of immune cells.

  1. Lines 261- 263. Replace “which promotes” to “by promoting”  to emphasize that the phosphorylation is promoted by LncDC and not by STAT3.

The following information is indicated in 363-366: Reports have indicated that lncDC promotes the phosphorylation of the transcription factor STAT3 (signal transducer and activator of transcription 3) at tyrosine-705 by preventing STAT3 from being dephosphorylated by SHP1.

  1. Could it be a typo by the authors and they meant shuttling and not shutting.

We apologize for this typographical error. Thank you for pointing out this mistake. Now the information is the following “The absence of the lncRNA Semaphorin 3B (SEMA3B) antisense RNA 1 (SEMA3B-AS1) causes nucleocytoplasmic shuttling of HMGB1 for release into the extracellular space”.

  1. Lines 361- 371. ER stressors and molecular mediators involved have already been presented before in this paragraph.

This paragraph indicates the role of some ER stressors in accumulation of unfolded or misfolded proteins and the regulation by some lncRNA.

  1. Two references of lncRNAs that could be important to this topic.

The information in these manuscripts is related to the involvement of the lncRNAs HAND2-AS1 and HOTTIP in the epithelial-mesenchymal transition, an event that is part of the hallmarks of cancer. However, we believe that the suggested information is not part of the cancer-immunity cycle concept originally proposed by Chen and Mellman (DOI: 10.1016/j.immuni.2013.07.012). Nevertheless, as suggested by reviewer 2, we include in the manuscript a section on the micropeptides produced by some lncRNAs associated with the cancer-immunity cycle. Section 4.6, lines 496- 542.

  1. Though proof-reading and English editing is required.

  1. Lines 456-458. HULC carcinoma epithelial de in….

We apologize for this mistake. This sentence was corrected, lines 425- 426.

  1. Line 487, The term “primordial” change to paramount, crucial, critical or essential.

Now this sentence was changed to: While the cells involved in the innate immune response are key for the early elimination of tumor cells, the ultimate goal of the cancer-immunity cycle is activation of the adaptive antitumor immune response.

  1. Line 509, the term microenvironment is somewhat ambiguous.

The immune cells require three signals, regardless of the environment in which they carry out their activation, so we consider it is not necessary to clarify this aspect.

  1. Information density and narrative slow should be greatly improved.

We hope that the manuscript in the submitted format will have a more organized and integrative flow of information and will be clearer and more interesting to readers.

  1. Critical citation and reference accuracy issues.

We apologize for these unacceptable mistakes. In this version, all the reference have been checked several times.

  1. Data presentation and organization.

Now, tables 1 and 2 contain information of all lncRNAs included in text.

Minor comments.

Front size in Fig.1 should be increased

Figure 2 has been added

  1. Homogenize terminology.

Abbreviations and terminology have been revised and standardized along the text.

Sincerely,

The authors

Reviewer 2 Report

Comments and Suggestions for Authors

I appreciate the authors for their well-structured and insightful manuscript, which provides valuable information on the role of long non-coding RNAs (lncRNAs) in cancer and immunity. Overall, this is a promising and timely manuscript. Addressing the suggestions would substantially enhance its clarity, scientific value, and utility for the field.

  • The manuscript is generally well-written. However, there are a few typographical errors that need correction. The overall use of English is appropriate.
  • Figure 1: The fonts are quite small and should be enlarged for better readability.
  • I recommend expanding the discussion to include other potential RNA-binding proteins involved in lncRNA-mediated immune regulation.
  • It would significantly improve the manuscript to add 1–2 more figures to support the sections: “Tumor Antigens: Release and Uptake”, “Regulating Phagocytosis in Cancer”
    Visual representation will enhance reader understanding of these complex processes.
  • The importance of transcriptomics in studying lncRNAs should be emphasized. Integrating a brief discussion of current diagnostic techniques (e.g., qRT-PCR, microarray, RNA-seq) would strengthen the clinical relevance of the manuscript.
  • Please include more detailed discussion on lncRNA-encoded micropeptides/proteins, their discovery, and their implications in cancer immunity. This is an emerging area that adds depth to the functional versatility of lncRNAs. May be a separate section.
  • It would be valuable to include a section on the turnover and stability of lncRNAs under normal vs cancerous conditions, and how this dynamic regulation affects their biological function.
  • I encourage the authors to add a brief discussion comparing the role of lncRNAs in early vs late-stage cancer diagnosis, highlighting their potential as non-invasive diagnostic biomarkers.
Comments on the Quality of English Language

The manuscript is generally well-written. The overall use of English is appropriate and fine in a general sense.

Author Response

Response to Reviewer 2, Manuscript IJMS 3513854

The authors are very grateful for the comments made on the manuscript, the suggested changes allow for a more organized and inclusive flow of information, and we hope that the text will be clearer and more interesting for its readers.

Front size in Fig.1 should be increased

Figure 2 has been added

We apologize for grammatical and typographical errors in the text. The manuscript was sent to an English proofreading service for correction.

The present version of the manuscript incorporates your innovative suggestion regarding lncRNA-encoded micropeptides. This information can be found in section 4.6, lines 496- 542.

The manuscript highlights information on the involvement of lncRNAs in promoting antitumor activities in immune cells and contrasts the action of lncRNAs favoring activities that promote tumor development and immune evasion mechanisms. Therefore, we did not consider including information on lncRNAs in tumor and non-tumor conditions and in early and late stages of cancer development. Although these topics are very important, they are outside our main proposal.

Best regards,

The authors

Round 2

Reviewer 1 Report

Comments and Suggestions for Authors

The authors have sincerely addressed most of my concerns. Upon rereading the manuscript I still have the following comments which I suggest improving for publication: 

1. Clarity of Conceptual Flow and Depth of Integration

While the manuscript has improved in organization, several sections still lack sufficient integration of mechanistic insight. For example, multiple lncRNAs are listed as being involved in the same pathway or phenotype (e.g., immune evasion or cytokine regulation), yet their comparative roles, and their potential redundancies or antagonisms are not discussed. A short interpretive summary or commentary at the end of key sections could greatly enhance reader understanding.

2. Figure and Table Utility

The addition of Figure 2 is appreciated. However, Table 1 and Table 2 have some redundancy in terms of both describing/listing lncRNAs that are affecting immune cells. Whereas in table 2 this is with a clear tumor perspecitve, I suggest the table titles may not clearly indicate the difference between the two tables. I suggest also clarifying that in text somehow. 

3. Balancing the "Therapeutic Discussion"

Section 9 on "therapeutic potential of lncRNAs" remains optimistic but would benefit from a more balanced discussion on translational limitations (e.g., in many cases poor conservation across species (though there are exceptions), delivery barriers, off-target effects - mentioned briefly but some more elaboration on this would be in place - just 1-2 sentences). Including these would increase scientific rigor and align the tone with current clinical development challenges.

Comments on the Quality of English Language

I still recommend thorough English editing prior to publication. 

Author Response

We sincerely thank the reviewer for their valuable feedback and for acknowledging the improvements made to the manuscript. Below, we provide detailed point-by-point responses to the remaining comments, along with a summary of the corresponding changes incorporated into the revised version.

We greatly appreciate the insightful comment regarding the need for deeper mechanistic integration. In response, we have incorporated interpretive summaries where appropriate—most notably in Section 6, where we now explicitly discuss the contrasting roles of MEG3 and TDRG1 in VEGF regulation and immune infiltration. These additions aim to enhance the reader’s understanding of the functional relationships among lncRNAs.

We acknowledge that further comparative analysis is still warranted, particularly in Sections 4 and 5.4, where multiple lncRNAs are involved in antigen presentation and T cell polarization. While comprehensive integration will require future expansion, we have taken initial steps in this version by highlighting shared pathways and outlining key functional contrasts between these lncRNAs.

We also thank the reviewer for the helpful suggestion regarding table clarity. In response, we have revised the titles of Table 1 and Table 2 to more accurately reflect their distinct scopes: Table 1 now focuses on the immunoregulatory roles of lncRNAs in immune cells, while Table 2 highlights their functions within the tumor microenvironment. Additionally, we have updated the manuscript text to clearly explain this distinction and guide the reader accordingly.

Regarding the therapeutic section, we agree that a more balanced discussion is essential. We have expanded Section 9 to address critical translational limitations, including:

  • the limited conservation of many lncRNAs across species, which complicates the use of preclinical models;

  • challenges associated with targeted delivery due to molecular size and tissue specificity; and

  • the potential for off-target effects.

These points have been integrated into the concluding paragraphs of Section 9 to align the discussion with current clinical development challenges.

In response to the recommendation for language improvement, we have carried out a thorough professional English-language revision of the entire manuscript. We are also submitting a certificate of editing as supporting documentation.

All modifications and newly added content in response to your comments are highlighted in yellow throughout the revised manuscript.

Once again, we sincerely thank the reviewer for their thoughtful and constructive input, which significantly contributed to enhancing the clarity, depth, and overall quality of our work.